# Intratumoral versus Circulating Lymphoid Cells as Predictive Biomarkers in Lung Cancer Patients Treated with Immune Checkpoint Inhibitors: Is the Easiest Path the Best One?

**DOI:** 10.3390/cells9061525

**Published:** 2020-06-22

**Authors:** Marta Gascón, Dolores Isla, Mara Cruellas, Eva M. Gálvez, Rodrigo Lastra, Maitane Ocáriz, José Ramón Paño, Ariel Ramírez, Andrea Sesma, Irene Torres-Ramón, Alfonso Yubero, Julián Pardo, Luis Martínez-Lostao

**Affiliations:** 1Medical Oncology Department, University Hospital Lozano Blesa, 50009 Zaragoza, Spain; lola.isla@gmail.com (D.I.); mara1290@gmail.com (M.C.); lastrarodrigo@gmail.com (R.L.); m.ocarizdiez@gmail.com (M.O.); andreasesmagoi@gmail.com (A.S.); irentorresr@gmail.com (I.T.-R.); ayuberoe@salud.aragon.es (A.Y.); 2Aragon Health Research Institute (IIS Aragón), 50009 Zaragoza, Spain; joserrapa@gmail.com (J.R.P.); pardojim@unizar.es (J.P.); lmartinezlos@salud.aragon.es (L.M.-L.); 3Carbochemical Institute (ICB-CSIC), Miguel Luesma 4, 50018 Zaragoza, Spain; eva@icb.csic.es; 4Infectious Disease Department, University Hospital Lozano Blesa, 50009 Zaragoza, Spain; 5Nanotoxicology and Immunotoxicology Unit (IIS Aragón), 50009 Zaragoza, Spain; aramirezlabrada@yahoo.es; 6ARAID Foundation (IIS Aragón), 50009 Zaragoza, Spain; 7Microbiology, Preventive Medicine and Public Health Department, Medicine, University of Zaragoza, 50009 Zaragoza, Spain; 8Biomedical Research Center in Bioengineering, Biomaterials and Nanomedicine Network (CIBER-BBN), 28029 Madrid, Spain; 9Immunology Department, University Hospital Lozano Blesa, 50009 Zaragoza, Spain; 10Department of Microbiology, Pediatrics, Radiology and Public Health, University of Zaragoza, 50009 Zaragoza, Spain; 11Aragon Nanoscience Institute, 50018 Zaragoza, Spain; 12Aragon Materials Science Institute, 50009 Zaragoza, Spain

**Keywords:** biomarkers, lung cancer, immune cells in cancer, tumor microenvironment, immune checkpoints inhibitors

## Abstract

The molecular and cell determinants that modulate immune checkpoint (ICI) efficacy in lung cancer are still not well understood. However, there is a necessity to select those patients that will most benefit from these new treatments. Recent studies suggest the presence and/or the relative balance of specific lymphoid cells in the tumor microenvironment (TEM) including the T cell (activated, memory, and regulatory) and NK cell (CD56dim/bright) subsets, and correlate with a better response to ICI. The analyses of these cell subsets in peripheral blood, as a more accessible and homogeneous sample, might facilitate clinical decisions concerning fast prediction of ICI efficacy. Despite recent studies suggesting that lymphoid circulating cells might correlate with ICI efficacy and toxicity, more analyses and investigation are required to confirm if circulating lymphoid cells are a relevant picture of the lung TME and could be instrumental as ICI response biomarkers. This short review is aimed to discuss the recent advances in this fast-growing field.

## 1. Introduction

Lung cancer is the most frequent tumor in the world, with more than 2 million new cases estimated in 2018 (11.6% of all cancers), and the one that causes the highest mortality (18.4% all cancers), according to the GLOBOCAN 2018 report [1].

Recently significant advances in the field of oncology, especially due to a better knowledge of tumor biology and the genomic profile of each patient, is allowing for the improvement of diagnosis accuracy as well as the incorporation of specific therapies against a variety of molecular targets. Specifically, the introduction of immunological therapies, immunotherapy, has been one of the greatest advances in lung cancer treatment of the last decades [2].

The study of the immune system has highlighted the importance of immune cells in the control of tumor growth, the so-called tumor immunosurveillance. During the immunological response, the first cells arriving to the tumor site are innate immune cells such as Natural Killer cells (NK) or Natural Killer T cells (NKT). NK cells are not only capable of killing tumor cells, but also promote the generation of cell debris that can be phagocyted by antigen-presenting cells (APC), mainly dendritic cells (DC). CD8^+^T lymphocytes interact with APC through their specific T cells receptors (TCR). Since every T cell clone expresses a unique TCR, the presence of millions of T lymphocytes in an individual leads to a large and highly variable TCR repertoire which ensures the recognition of a plethora of foreign antigens, including tumor mutated antigens. This cell-to-cell contact in the immunological synapse promotes the conversion of naïve CD8^+^T lymphocytes into specific antigen-activated cytotoxic T lymphocytes (Tc) capable of eliminating tumor cells. This is achieved by inducing the expression in these T cells of cytotoxic molecules, such as perforin or granzyme B.

Under some circumstances, tumor cells are able to overcome the host immune response by acquiring immune evasion mechanism such as inhibitory molecules that, under physiological conditions, are involved in immune response homeostasis and tolerance. T lymphocytes express inhibiting receptors involved in tolerance regulation against self-antigens, such as CTLA-4 (Cytotoxic T Lymphocyte Antigen–4) and PD-1 (Programmed Death–1), which are used by tumor cells to create an immunosuppressive state and escape of the immune system. The union between CTLA-4 and B7-1 (CD80) or B7-2 (CD86), expressed in the antigen-presenting cells, and the interaction between PD-1 and its ligands, PD-L1 or PD-L2 (Programmed Death Ligand-1,-2), commonly expressed by tumor and other cells in the tumor microenvironment (TME), inhibits T cell response. In addition, other effects of PD-1/PD-L1 signaling on other lymphocytes have been described, including the increase of proliferation of T regulatory cells (Treg), enhancing the immune-suppressive environment as well as the decrease in the activity of both effector Tc and NK cells [3,4,5].

In recent years, the most relevant immunotherapeutic treatments have been aiming to block PD-1/PD-L1 interaction using anti-PD-1, antiPD-L1, and/or anti-CTLA-4 monoclonal antibodies. Despite the enormous potential of immunotherapy, not all patients or all tumors respond in the same manner to treatment. There are still difficulties to know the best way to select patients that will greatly benefit from this type of therapy, and the only biomarker with clinical utility for immunotherapy, such the expression of PD-L1, which has been studied in many clinical trials [6,7,8], is not completely accurate and presents several limitations [9,10,11].

Responding to this challenge, several studies have been carried out in order to find predictive response biomarkers to anticipate which patients will respond to each type of treatment, avoid unnecessary toxicities, and reduce costs. [2]. In this regard, a wide range of studies has explored the possibility that the tumor mutational burden, the composition of the gut microbiome, and the profile of immune cell populations can be useful predictive biomarkers of response [12].

In this review we are going to focus on the experimental evidences of how different immune cells types present in the TME and their distribution in peripheral blood can contribute to predict a good response to immunotherapy against lung cancer. In this context, our aim from a translational point of view is to review the most important studies evaluating immune cells as predictive biomarkers of response in lung cancer making special emphasis on the advantages and potential limitations of using samples obtained through minimal invasive procedures such as peripheral blood samples.

## 2. Tumor Microenvironment

A wide range of evidence has demonstrated that TME is essentially involved in the immune response. The components of this microenvironment include macrophages, T cells, B cells, dendritic cells, NK cells, myeloid-derived suppressor cells (MDSCs), and neutrophils, which represent a potentially powerful prognostic instrument for lung cancer. All these components create, by different mechanisms, an immunosuppressive state that allows tumor cells to remain hidden from the immune system and to continue their growth. Some of these immune-escape mechanisms are an inappropriate presentation of tumor antigens, the secretion of inhibitory cytokines and chemokines, tumor mutations to achieve molecules unrecognizable by the immune system, and the attraction and recruitment of inhibitory cells [13].

One of the most important factors implicated in the modulation of cancer immunity in TME are TILs (tumor-infiltrating lymphocytes), especially cytotoxic T cells, memory T cells, and helper T cells [14].

Due to the role of T regulatory lymphocytes (Treg), they seem to have an important role in immunosuppression. The activation of CTLA-4 in Treg upregulates the expression of FOXP3 (Forkhead box P3) which is the central transcription factor that regulates the development and function of CD4^+^ Tregs. In addition, this activation involves the conversion of peripheral CD4^+^ CD25^-^ naive T cells into CD4^+^ CD25^+^ regulatory T cells. Additionally, PD-1 expression in Treg stimulates their proliferation, contributing all of this to immunosuppressive state and cancer proliferation [13].

### 2.1. TILS and CD8^+^ T Cells

Infiltration of immune cells in TME and its relation in response to immunotherapy has been analyzed in recent works. Multiple studies have shown the importance of TILs and how they correlate with tumor elimination and better responses to anti-immune checkpoint based therapies. TILs are considered a selected population of T cells with a higher specific immunological reactivity against tumor cells than non-infiltrating lymphocytes. Among TILs, neoantigen-specific activated CD8^+^ T cells are one of the most relevant cell populations since they have the potential to recognize and eliminate cancer cells [15].

Tumors with a substantial amount of DNA damage that might originate neoantigens commonly induce a specific reactive CD8^+^ T cell response. However, concomitantly with the generation of anti-tumor activated CD8^+^ T cells, DNA damage also upregulates some immunosuppressive mechanisms within TME, such as the expression of the immune checkpoints PD-1, CTLA-4, Tim-3 and LAG-3 on TILs. The interaction of these receptors with their corresponding ligands expressed on other cells is associated with an impaired proliferation and effector function, frequently referred to as exhaustion. PD-1 / PD-L1 inhibitors block the negative regulatory signal pathway and unleash T cells from exhausted status. Therefore, it is not surprising that a higher expression of PD-1 and CTLA-4 in lymphocytes correlates with a better response to immunotherapy [16].

In this line, Thommen and coworkers described the properties of T cells with different levels of PD-1 expression in non-small cell lung cancer (NSCLC) and their potential relationship with immune-mediated tumor control [16]. Their data showed that CD8^+^ TILs with levels of PD-1 that exceed those commonly observed on healthy controls, present characteristics that suggest they are one of the most important cells involved in tumor recognition. These T cells display molecular, metabolic, and functional differences relative to those with a PD-1 normal or negative expression. PD-1^+ high^ CD8^+^ T cells show an increased expression of other inhibitory receptors as Tim-3, LAG-3, TIGIT, CD244, and BTLA. Moreover, they failed to produce substantial levels of IL-2, TNF-α, IL-10, and IFN-γ, and differ in the expression of CD28. Regarding TCRβ analysis, PD-1^+ high^ CD8^+^ T cells are characterized by significantly higher clonality, with the top 30 clones contributing close to 90% of the entire TCR repertoire. In addition, several genes involved in cell cycle regulation and proliferation were upregulated with respect to CD8^+^ T cells with a PD-1 normal or negative expression. Concerning their metabolic status, PD-1^+ high^ CD8^+^ T cells showed an upregulation of glycolysis and lipid uptake also showing an increased number in mitochondria. However, these mitochondria were dysfunctional and presented reduced mitochondrial membrane potential. Finally, PD-1^+ high^ CD8^+^ T cells upregulate the expression of CXCL13, which function is to attract other immune cells subsets to the TME. Based on this, PD-1^+ high^ CD8^+^ T cells may reflect a direct tumoricidal capacity and an ability to recruit other immune cells that mediate the antitumor immune response, following a PD-1 blockade treatment. Based on these data, authors did a retrospective analysis of pretreatment biopsies of 21 stage IV NSCLC patients undergoing anti-PD-1 therapy, revealing a clear correlation between the presence of these PD-1^+ high^ CD8^+^ T cells and the response to treatment and also with the overall survival (OS) (HR 0.16, *p* < 0.05) [15,16,17].

In this context, there are some clinical trials exploring this possibility with promising results. It has been noticed that the presence before treatment of high percentage of PD-1^+^ CD8^+^ TILs in NSCLC treated with durvalumab had better overall response rate (ORR) (37% vs. 7%), better OS (24.3 vs. 6.5m), and also better progression-free survival (PFS) (7.3 vs. 2.6m) compared to those patients with a low percentage of PD-1^+^ CD8^+^ TILs [18]. Another clinical trial with atezolizumab in NSCLC obtained similar results with a better ORR [43% vs. 8%] and PFS [6.8% vs. 2.8%] in patients with high presence of PD-1^+^ CD8^+^ cells in TME [19].

All these studies confirm and expand to human trials, previous findings in other types of tumors, such as melanoma, showed PD-1 expression in CD8^+^ TILs in tumor sample defines clonally expanded tumor-reactive lymphocytes [20].

### 2.2. CD8/CD4 Ratio

Based on their function CD4^+^ T lymphocytes are classified in two main subsets, T helper lymphocytes (Th) with a co-operative activity, and Treg which participate in the immune regulation, avoiding an excessive immune response. It has been extensively shown that Treg cells inhibit the antitumoral function of DC, NK, or CD8^+^ T cells, among other means by expressing PD-L1 [21]. In this context it has been recently shown that NSCLC patients with a high frequency of intratumor PD-1^+ high^ CD8^+^ T cells and PD-L1^+ high^ CD4^+^ Treg present a better clinical response during anti-PD-1 treatment [22,23].

However, it should be noted that according to a meta-analysis study performed in 2011, the quantitative ratio between different immune cell populations in TME could be more significant than their mere presence [24,25]. Indeed, a high CD4^+^ Tregs / CD8^+^ T cytotoxic cell ratio was found to indicate a bad prognosis in NSCLC patients [26,27]. The correlation of this ratio with the response against ICI (immune checkpoint inhibitors) has to be confirmed in NSCLC patients.

Here, it should be pointed out that within the Tregs cell subsets, different markers have been used to characterize their phenotype and functionality. As previously explained, FOXP3 is the central transcription factor that regulates the development and function of CD4^+^ Tregs. However, the expression of FOXP3 in T cells and its impact is still controversial. Some studies have shown that FOXP3 expression is observed in activated T cells with no regulatory activities, whereas other ones indicate that it is mostly associated with T cells with regulatory activities [28,29]. There is accumulating evidence that FOXP3^+^ T cells are heterogeneous in phenotype and function, consisting of both suppressive and non-suppressive subpopulations. If only FOXP3 is measured to differentiate Treg and define them as T cells with regulatory activity, it will run the risk of being improperly classifying them, selecting some FOXP3 ^+^ T cells with cooperative activity. This can lead to contradictory results when evaluating these cells as markers of response to immunotherapy. In fact, FOXP3^+^ CD4^+^ T cells can be divided into three subpopulations based on the expression levels of FOXP3 and CD45RA: (i) FOXP3^low^ CD45RA^+^ CD25^low^ cells, designated as naive or resting Treg cells; (ii) FOXP3^high^ CD45RA^−^ CD25^high^ cells, designated as an effector or activated Treg cells; and, (iii) FOXP3^low^ CD45RA^−^ CD25^low^ non-Treg cells, which do not possess suppressive activity but can secrete pro-inflammatory cytokines. Regarding this functional characterization, it was recently found that the subset of Treg cells with higher immunosuppressive activity is characterized as CD25^high^CD127^low^which might facilitate its characterization in patient-derived samples. Furthermore, populations of highly suppressive group ii (FOXP3^high^ CD45RA^−^ CD25^high^ cells) and non-suppressive group iii (FOXP3^low^ CD45RA^−^ CD25^low^ cells) can be better differentiated by the expression of CD15s (sialyl Lewis x), a sugar antigen present on suppressive Treg cells, at least for those located in peripheral blood (30). Therefore, it is crucial to assess heterogeneity of FOXP3^+^ T cells in tumor tissues in order to evaluate their contribution to anti-tumor immune response. In this line, accumulating studies have demonstrated that a large number of Treg cells and a decreased ratio of tumor-infiltrating CD8^+^ T cells *versus* FOXP3^+^ Treg cells were shown to correlate with poor prognosis in different types of cancer including NSCLC [27,29,30].

In this context, as mentioned above, some studies are trying to clarify the role of the CD8^+^/CD4^+^ ratio as a biomarker response of immune checkpoint-based therapies. Uryvaev and colleagues observed that patients with a ratio of CD8^+^/CD4^+^ in TME lower than 2 had a response rate of 13.3% (*p*: 0.046) in metastatic NSCLC treated with anti-PD-1, whereas if it is higher than 2, the response rate ascends to 43–50% (*p*: 0.038). [31].

### 2.3. T Cell Memory

Immune response mediated by T lymphocytes against an antigen usually results in the generation of a persistent population of antigen-experienced cells that represent immunologic memory. Therefore, it is expected that antigenic tumors such as melanoma or NSCLC that have been shown to trigger a host immune response during tumor development and progression also generate immune memory against specific antigens. Both effector and memory T cells are restricted by PD-1-mediated inhibition, which partly explains the ongoing ineffective immune response in this type of patient. In this context, some studies have been designed to explore the possibility that the proportion of memory T cells could have a correlation with the immunotherapy response.

Exploring the different types of T cells, Ribas and colleagues carried out an examination of biopsies from melanoma samples during pembrolizumab treatment [32] finding an increase in the frequency of memory T cells (CD8^+^/CD4^–^/CD45RO^+^) in patients with a good response to pembrolizumab.

In this line, a novel population of memory CD8^+^ T cells called resident memory T cells (Trm) were identified and could play a key role in tumor immunosurveillance and ICI immunotherapy [33,34]. These cells are characterized by staying on their local tissue without recirculating in the blood. This subset is defined by expression of CD103^+^, CD49a^+^, and CD69^+^ and it has been observed in lung cancer TILs as well as other tumors [35,36].

Among their properties, Trm cells respond faster to re-exposure to tumor antigen than circulating memory cells and express high levels of cytotoxic molecules [34]. Trm cells may be functionally exhausted within the TME by the induction of T-cell inhibitory receptors including PD-1 and Tim-3 and preliminary data showed that they could expand early during anti-PD-1 treatment, being, in fact, a promising biomarker [37]. Recently, Clarke J. et colleagues studied this Trm in lung cancer patients and identified a Tim-3^+^ IL-7R^−^ Trm subset present exclusively in tumors which expressed high levels of PD-1. These cells may be one of the cellular targets of anti-PD-1 therapy and differences in the magnitude of this population of Trm could be an explanation for the variation in the clinical response to PD-1 inhibitors [38].

In this sense, another study demonstrated that a high infiltration of CD103^+^ CD8^+^ Trm TIL subset in lung cancer correlates with improved early-stage NSCLC patient survival [39]. Moreover, this study showed a positive correlation between dense tumor infiltration by CD103^+^ TIL, irrespectively of epithelial and/or stromal location and NSCLC patient PFS. Interestingly, it has proposed that since CD103^+^ CD8^+^ TIL frequently express PD-1 and Tim-3, and this T cell subset is much more highly susceptible to activation-induced cell death than CD103^-^ T cell subset, targeting PD-1/PD-L1 signaling could enhance the anti-tumor activity of this population. Therefore, the measurement of this T cell subpopulation in TME could be an interesting biomarker of a good outcome.

### 2.4. Natural Killer Cells

Among the multiple mechanisms that cancer uses to escape from the immune system, the loose expression of the Mayor Histocompatibility Complex (MHC) is a common mechanism to avoid T cell-mediated destruction [40]. High levels of PD-L1 expression have been observed in tumors with low class I MHC expression but surprisingly some of these cancer types are responsive to PD-1/PD-L1 blockade. These findings suggest the existence of an immune response independent of T cells, inhibited by PD-1, and rescued by PD-1 blockade, which was recently identified as depends on NK cells during cancer [41].

NK cells are innate lymphocytes with cytotoxic activity against cancer cells and their presence in solid tumors, as lung cancer, is a good prognostic factor and improves OS [42,43,44,45,46,47,48,49]. Unlike T cells, recognition of tumor cells by NK cells does not require antigen presentation by HLA [50], and, in contrast with T cells, NK cells are more active against tumor cells with low MHC expression [51].

Joy Hsu and coworkers observed that T and NK cells cooperated during treatment with anti-PD-L1 in several mouse models including lung cancer models. On the one hand, NK cells supported T lymphocytes in the tumor response against tumors with high expression of MHC and self-antigens, and on the other hand, they became the main cytotoxic cells in those tumors with low expression of MHC even in PD-L1 negative tumors [41].

As with T cells, NK cells could express PD-1 in cancer patients, regulating its activity against tumors and contributing to PD-1 based immunotherapy. Only a small fraction of NK cells expresses high levels of PD-1 corresponding to a fully maturated adaptive NK cell subset expressing CD57 and NKG2C [50]. However, it has been found that its antitumoral function can be partly restored by anti-PD-1/L1 antibodies [52].

Other data suggest that in cancer patients, NK cells are potentially modulated by PD-1 blockade [41,50,53]. However, despite NK cell infiltration is a good prognosis biomarker in different cancers including SCLC and NSCLC [43,47], its value to predict ICI efficacy is still not clear. However, as discussed below new studies indicate a correlation between circulating NK cells and antiPD-1/-L1 efficacy.

### 2.5. Immune Prognostic Groups

The increasing evidence of the relationship between the immune system and tumor growth has created the need of dividing tumors not only by size and metastatic affectation, but also by phenotype, function, and location of immune cells.

Recently immune TME model has been established as an immunotherapy predictor, distributing tumors in four groups depending on the expression of PD-L1 (positive or negative) and TIL status (positive or negative). According to this classification, a tumor with both positive biomarkers is going to achieve an adequate response to immunotherapy. However, if TME is PD-L1^+^ but present a negative TIL status, it is more likely to be refractory to PD-1/PD-L1 monotherapy [54].

This model has been used in different studies to divide tumors based on host immunity-tumor interactions and four phenotypes have been proposed: (i) hot, (ii) excluded, (iii) immunosuppressed, and (iv) cold. Hot immune tumors show a high degree of T cell infiltration and high immune checkpoint expression (PD-1, CTLA-4, Tim-3, and LAG-3). In contrast, cold tumors are characterized for an absence of T cells within the tumor and, thus, do not respond to ICI (low tumor mutational burden, poor antigen presentation, and intrinsic insensibility to T cell-mediated killing). The excluded phenotype reflects the tumor‘s ability to remain hidden from the immune system with an absence of T cell infiltration inside the tumor bed and accumulation at tumor surrounding tissue, frequently blocked by physical barriers such as extracellular matrix or fibroblasts. Finally, immunosuppressed phenotype shows a low degree of tumor infiltration due to an immunosuppressive environment in the tumor with the presence of immunosuppressive cells and soluble inhibitory mediators [55]. In patients with metastatic melanoma, the presence of immunosuppressed, excluded, and cold phenotypes correlate with worse responses to both antiCTLA-4 and antiPD-L1 blocking antibodies than hot tumors [56,57]. Recently, Lizotte and coworkers identified these phenotypes also in NSCLC [58]. It is expected that a more detailed study about the factors modulating tumor immunity in TME will improve these models, creating individualized TME immune fingerprints to guide personalized treatments and to improve ICI efficacy by developing new drugs or combinations.

## 3. Peripheral Blood

In many cases of lung cancer, it is difficult to obtain enough biopsy tissue to reflect all tumor heterogeneity and to properly study TIL phenotype and location in situ. Thus, increasing studies have focused on studying the profile of blood circulating immune cells to predict the immunotherapy response as these samples are easier to obtain than lung biopsies. Analysis of lymphocyte populations in peripheral blood is a low invasive method with interesting potential to predict the treatment outcome after immune-based therapies.

Although immune cells present in blood might not be restricted to some of the components of the TME, its presence might indirectly reflect the regulators present in TME. Indeed, some studies have correlated different inflammatory biomarkers, such as T cells, DC or NK cells, in blood and tumor tissue samples and found a good correlation between both sites [59,60].

Regarding T cells, it was found that patients with a good response to ICIs tend to have a decreased number of T cells in peripheral blood at baseline in comparison with non-responding ones [61]. Complementing this study, Kamphorst and coworkers observed that an early expansion of peripheral PD-1^+^ CD8^+^ T cells during anti PD-1 treatment in NSCLC had a positive effect in treatment response. Whereas 70% of patients with disease progression had a delayed or absent PD-1^+^ CD8^+^ T-cell response, 80% of patients with clinical benefit exhibited PD-1^+^ CD8^+^ T-cell responses within 4 weeks of treatment initiation [62]. This result was confirmed by Mazzaschi and colleagues in NSCLC patients treated with nivolumab [63].

Another study in NSCLC patients treated with nivolumab showed the CD4^+^ and CD8^+^ central memory T cell (CM) / Effector T cell (Eff) ratio in blood is useful to correlate with treatment response to anti-PD-1 immunotherapy. The data showed that high CM/Eff T cell ratio correlated with increased tumor PD-L1 expression and longer PFS (91 vs. 215 days). This apparently non-expected result, inversely correlating Eff T cell number and ICI response could reflect the presence of terminally differentiated T cells that are unable to reach and eliminate the tumor [64]. More recently it has been shown that longer OS in patients treated with nivolumab had higher levels of CD3^+^, CD4^+^, and CD8^+^ T cells but lower levels of NK cells at baseline.

Regarding NK cell populations, a few studies have analyzed the relationship between circulating NK cells and their phenotype with prognosis and response to ICI with some surprising unexpected results that will require further clarification and confirmation [65]. Circulating NK cells are mainly classified in the basis to the level of CD56 and CD16 expression as CD56^dim^CD16^+^ and CD56^brigh^CD16^low/-^ cells, presenting cytotoxic or immunoregulatory function respectively [66].

Mazzaschi and coworkers studied the relationship between NK and Treg cells, observing that, at a baseline, CD 56^+^ NK cells resulted in 2-fold higher in the responder group compared to non- responders in NSCLC. In addition, following PD-1 blockade, the increase in the number of NK cells in responder patients was accompanied by a significant increase in the number and proliferation of Tregs. Conversely, in patients who did not respond to immunotherapy the number of circulating Tregs and NK cells progressively declined [63]. Intriguingly, as indicated above, it has been found that low blood NK cell counts at baseline are a good predictive biomarker for good response in patients treated with nivolumab [67]. However, it should be noted that authors did not differentiate between CD56^bright^ non-cytotoxic and CD56^dim^ cytotoxic NK cell subsets and, thus, results are not conclusive regarding NK cells. Indeed, it was previously found in melanoma patients that circulating CD56^bright^ NK cells inversely correlated with survival of melanoma patients [68] and that tissue from NSCLC is infiltrated with CD56^bright^ non-cytotoxic NK cells [69].

Last but not least, it was found that the level of circulating NKp46^+^CD56^dim^ NK cells inversely correlated with prognosis in NSCLC although it was not analyzed the response to ICI [70].

As previously mentioned for the TME cells subset, it has also been found that the ratio between circulating lymphocytes, would be a more specific way of predicting the response to immunotherapy in peripheral blood. In this context, it has been observed a relationship between the effector and suppressor cell ratio and the response to nivolumab treatment in NSCLC with a significantly higher value in CD 56^+^ NK cells + PD-1^+^ CD8^+^ T cells / MDSCs + FOXP3^+^ Tregs ratio in responders than no responders (7.41 vs. 4.64) [63].

Taking into account the CD4^+^/CD8^+^ ratio, it has been shown that a high CD8^+^/CD4^+^ baseline ratio in peripheral blood statistically significantly correlates with better prognosis in early-stage NSCLC (HR 0.19) [71].

Other biomarkers explored based on lymphoid cells of peripheral blood are the neutrophil to lymphocyte ratio (NLR), and absolute lymphocyte count (ALC). In this regard, Zer and coworkers explored the potential of these counts as biomarkers of response to ICI in NSCLC. Regarding NLR, patients with a baseline < 4 showed a better disease control, longer treatment duration, longer time to progression, and increased OS (p:0.54). Moreover, NLR < 4 at week 8 of treatment is associated with objective response to treatment (p: 0.025).

This prognostic potential of decreased NRL likely reflects increased lymphocyte-dependent inflammation as well as reduced neutrophil activity, both of which can lead to tumor regression and better prognosis [72]. These data were also studied in other tumors such as colorectal cancer treated with chemotherapy and renal cancer treated with sunitinib with similar results [73,74,75].

The role of ALC as a predictive biomarker has been confirmed in metastatic melanoma patients treated with ipilimumab. Patients with 1.35-fold higher ALC values from baseline in the first 2 weeks of treatment had significantly higher OS [22,76].

Adding a layer of complexity, Pryyanka and colleagues found that anti-CTLA-4 and anti-PDL-1 ICI have distinct predictive biomarkers in circulating lymphocytes. On the one hand, patients with a lower number of CD45RA^+^ T cells in both CD4^+^ T cells (p: 0.038) and CD8^+^ T cells (p: 0.019) presented worse response to anti-CTLA-4. Additionally, memory T cells were higher in responders compared to no responders. In contrast, these correlations were not found in antiPD-L1 treated patients [77]. In this case, responding patients had significantly higher numbers of CD69^+^ CD56^dim^ NK cells than no responding ones, not observing any differences in anti-CTLA-4 patients. Surprisingly, authors also observed an association between higher expression of the cytotoxic protein granzyme B in NK cells and worse response in anti-PD-L1 patients [77]. This result will require further clarification since granzyme B is a marker of functionally competent NK cells. Another study, also found in NSCLC patients treated with nivolumab that the number of circulating CD 56^+^ NK cells were 2-fold-higher in responders compared to no responders [63].

Unfortunately, the number of studies is limited and new works will be required to confirm and expand the potential of circulating T and NK cell subsets as predictive biomarkers of ICI response [10,12].

Finally, another important aspect, and one of the most important adverse outcome of ICI treatment in lung cancer, is the Hyperprogressive disease (HPD). This consists of an unexpected acceleration of tumor growth associated with a premature clinical deterioration [78]. Several studies have been carried out with the aim of understanding the underlying mechanisms and identified biomarkers. A study with 70 NSCLC patients treated with ICI, demonstrated that HPD is correlated with a dysfunctional CD4 response and an increase in CD4^+^ CD28^-^ in peripheral blood [79]. In addition, another study with 253 NSCLC patients correlated HPD with a lower frequency of effector/memory subsets (CCR7^-^ CD45RA^-^ T cells among the total CD8^+^ T cells) and a higher exhausted population of T cells [80].

Table 1 includes all the studies that have been carried out in order to confirm the utility of these biomarkers in cancer patients treated with immunotherapy.

## 4. Concluding Remarks

The increasing knowledge of the regulation of cancer immunity by immune checkpoints is revealing the complexity of host immunity-cancer interaction. Thus, although the emergence of ICIs has allowed treating some aggressive cancers, still the number of patients benefiting is relatively low, and good predictive biomarkers to stratify patients according to expected responses are required to increase ICI efficacy and develop new drugs and combination.

Discovering biomarkers of response is one of the current challenges in oncology and many studies are currently underway trying to find specific biomarkers of response that could help in the clinical practice avoiding toxicities and selecting patients for the more effective treatment.

As summarized in this review, lymphoid cells could be a good biomarker of response both in TME and peripheral blood. However, studies in TME are limited due to the difficulties to get a representative tissue sample in lung cancer biopsies. Thus, more studies in peripheral blood are required to confirm the initial analyses suggesting a correlation between specific T cell and NK cell subsets and ICI response as well as to reveal new cell-based biomarkers. Specifically, since tools for analyzing antigen-specific T cells (i.e., tetramers) in lung cancer are limited, the study of tumor-specific reactive T cells as biomarkers has not been afforded as it was done in melanoma patients [82]. In this sense, different approximations that still need validation have been initiated. Identification of activated T cells that might have contacted tumors characterized by circulating neoantigen-specific granzyme B^+^ PD-1^+^ T cells in melanoma [82] and a similar pool in lung cancer albeit antigen-specificity was not assessed in this case [62]. The second is the identification of T cell clonal expansion and variability TCRβ sequencing [83].

TCRβ sequencing could be used as a tool to identify specific T cell populations for tumor neoantigens that could be useful as biomarkers for a good response to ICI. This is suggested by previous results in melanoma, indicating that pre-existing antigen-specific PD-1^+^ CD8^+^ T cells in the blood are functional activated cells that suffer from exhaustion after interaction with cancer cells [80]. Thus it should be expected that patients presenting a high number of clonally expanded T cells presenting a high variability of TCRs, meaning high numbers of potentially tumor-specific T cells, should present good responses to ICI as previously found in melanoma and pancreas cancer patients treated with anti CTLA-4 [84,85,86].

The number of studies analyzing TCRβ as a predictive biomarker in lung cancer is scarce. A recent study has analyzed its prognostic significance [87] and just the first study in NSCLC recently indicated a correlation between TCRβ diversity and anti PD-1 response [81]. Importantly the predictive value of TCRβ sequencing was restricted to PD1^+^ CD8^+^ T cells and was loss when total PBLs were used. More studies will be required to validate these findings and to show the potential of this biomarker to predict ICI response in lung cancer.

In addition, as above explained, different evidence indicates that cell biomarkers might differ depending on the ICI treatment and, thus, it seems that specific biomarkers for the different immunotherapeutic agents currently used will be required. It is not difficult to understand that probably not all biomarkers will be useful for all types of tumors and all types of immunotherapy treatments, so one more time personalized approach will be required to optimize ICI efficacy and to reduce toxicity.

Another aspect to consider is the possible differences existing in elderly patients due to immunosenescence. With age, immune system function is impaired and disrupts the expression of PD-1 and CTLA-4, so the ability to respond to immunotherapy may be altered. Other important age-related changes are: (i) a reduction in CD8^+^ naïve T cell population; (ii) a stable proportion of CD4^+^ naïve T cells but with an altered function; (iii) a rise of memory CD4^+^ T cells and (iv) a higher concentration of inflammatory cytokines [88]. In this context, Kornelis and colleagues studied CD 161^+^ (C-type lectin receptor) T cells population, which identifies subsets of CD4^+^ and CD8^+^ T cells with a strong pro-inflammatory phenotype, in peripheral blood of elderly patients. They show that aging is associated with a numerical decline of circulating CD161^high^ CD8^+^ T cells, as well as a stable number and function of CD161^int^ CD8^+^ T cells and a decreased production of pro-inflammatory cytokines by CD161^+^ CD4^+^ T cells [89]. For these reasons, there are some limitations in terms of biomarkers of response in this cluster of patients, and more studies should be done in elderly patients.

Although increased evidence points to circulating lymphoid cells are useful biomarkers to predict ICI response, still more analyses and investigation will be required in lung cancer patients treated with immunotherapy, to validate lymphoid cells as biomarkers of response during ICI treatments and to exploit all the potential of the immune system to fight this still highly fatal disease. The previously described biomarkers related to response and non-response to immunotherapy in TME and peripheral blood are summarized in Figure 1.

## Figures and Tables

**Figure 1 cells-09-01525-f001:**
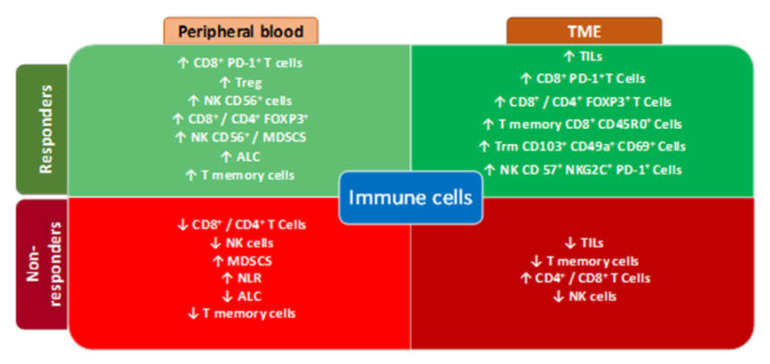
Biomarkers of response in TME and Peripheral blood. PD-1: Programmed Death – 1, TME: tumor microenvironment Treg: T regulatory cell, FOXP3+: Forkhead box P3, NK: Natural killers, MDSCS: myeloid-derived suppressor cells, ALC: absolute lymphocyte count, TILs: tumor infiltrating lymphocites, Trm: resident memory T cells, NLR: neutrophil to lymphocyte ratio.

**Table 1 cells-09-01525-t001:** Studies from biomarkers of response.

Tumour	ICI Treatment	Type of STUDY	Findings associated with clinical response	Ref.
NSCLC	anti PD-L1	Retrospective	PD-1+ CD8+ T cells in TME associated with treatment response and better OS	[16]
NSCLC	durvalumab	Clinical Trial	Pretreatment PD-1+high CD8+ T cells in TME associated with better ORR, OS and PFS	[18]
NSCLC	atezolizumab	Clinical Trial	Pretreatment PD-1+high CD8+ T cells in TME associated with better ORR and PFS	[19]
NSCLC	pembrolizumab nivolumab	Prospective	PD-L1+high CD8+ and PD-1+high CD4+ Treg cells in TME associated with better clinical outcome	[23]
NSCLC	anti PD-L1	Retrospective	CD8+/CD4+ ratio >2 in TME associated with better response	[31]
Melanoma	pembrolizumab	Clinical Trial	High T memory cells CD8+/CD4-/CD45R0 in TME in responder patients	[32]
NSCLC	Immunotherapy	Prospective	High CD103+ CD8+ Trm cells in TME in early NSCLC patients associated with better OS Positive correlation between dense tumor infiltration by CD103+ TIL and NSCLC patient PFS	[39]
NSCLC	pembrolizumab nivolumab atezolizumab	Prospective	Early on treatment proliferative PD-1+ CD8+ T cells in peripheral blood associated with a positive effect in treatment response	[62]
NSCLC	nivolumab	Prospective	Early on treatment proliferative PD-1+ CD8+ T cells in peripheral blood associated with a positive effect in treatment response CD56+ NK cells 2 fold higher in the responder group in peripheral blood High Treg in the responder group in peripheral blood Higher CD 56+ NK cells + PD-1+ CD8+ T cells / MDSCs+ FOXP3+ Tregs ratio in responders in peripheral blood	[63]
NSCLC	nivolumab	Retrospective	Longer PFS in patients with high T cell CM/effector ratio in peripheral blood.	[64]
NSCLC	nivolumab	Prospective	Higher levels of CD3+, CD4+, and CD8+ T cells correlate with longer OS Lower levels of NK cells at baseline correlate with longer OS	[67]
NSCLC	Immunotherapy	Prospective	The overall activity or number of NK cells is elevated in the immunotherapy responder group when compared with non-responders	[65]
NSCLC	Immunotherapy	Cases-control	High CD8/CD4 baseline ratio in peripheral blood associated with a better prognosis	[71]
NSCLC	anti PD-L1	Retrospective	Baseline NLR < 4 in peripheral blood associated with better disease control, longer treatment duration, PFS and OS NLR < 4 at week 8 of treatment in peripheral blood associated with objective response	[72]
Melanoma	ipilimumab	Prospective	1.35 fold-higher ALC from baseline in peripheral blood in the first 2 weeks associated with better OS	[76]
Melanoma	anti PD-L1 anti CTLA-4	Retrospective	Lower number of CD45RA+ T cells in both CD4+ and CD8+ T cells in peripheral blood in anti-CTLA-4 responder patients T memory cells higher in responders compared to no responders to anti CTLA-4 patients in peripheral blood Higher number of CD69+ NK cells in peripheral blood in responders than no responders to anti PD-L1 An association between higher expression of granzyme B in NK cells in peripheral blood and worse response in anti-PD-L1 patients	[77]
NSCLC	anti PD-1	Cohorts	High PD-1+ CD8+ TCR diversity pretreatment showed better ORR and PFS to ICI compared to patients with low diversity	[81]

**NSCLC:** no small cell lung cancer, **CTLA-4**: Cytotoxic T Lymphocyte Antigen – 4, **PD-1**: Programmed Death – 1, **TME:** tumor microenvironment **Treg**: T regulatory cell, **FOXP3+**: Forkhead box P3, **NK**: Natural killers, **MDSCS**: myeloid-derived suppressor cells, **ALC**: absolute lymphocyte count, **TILs**: tumor infiltrating lymphocites, **Trm**: resident memory T cells, **NLR**: neutrophil to lymphocyte ratio. **OS**: Overall survival **PFS**: progression free survival, **ORR**: overall response rate, **CM**: central memory.

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
