# Peer review of "Intratumoral versus Circulating Lymphoid Cells as Predictive Biomarkers in Lung Cancer Patients Treated with Immune Checkpoint Inhibitors: Is the Easiest Path the Best One?"

_cells, 2020, doi:10.3390/cells9061525_

Round 1

Reviewer 1 Report

The topic of the manuscript is of high relevance in the current scenario as there is urgent need of appropriate response biomarkers. The authors focused on the different immune cells presents in TME and on their peripheral distribution. The paper appears well written.

Comments:

Authors did this review in order to

The structure of the manuscript is good.

The table is well done as well as figure has charm and is of sufficient clarity.

However, few points need to be fixed.

Line 118: No small cell should be non-small cell lung cancer.  

128 – 132: these line appears quite confusing, please fix them.

In the background TME please discuss data from this early review which highlighted the network between innate immune cells and cancer cells

https://www.ncbi.nlm.nih.gov/pmc/articles/PMC6801651/

https://www.sciencedirect.com/science/article/abs/pii/S1471489218300031?via%3Dihub

In the conclusion , please introduce the concept that these findings might be different in special cluster of patients as elderly due to immunosenescence and  inflammaging which may alter the circulating lymphoid cells. Please include other limitations in this field of research.

https://www.ncbi.nlm.nih.gov/pmc/articles/PMC5917671/

https://www.ncbi.nlm.nih.gov/pmc/articles/PMC6539213/

Author Response

Thank you very much for the review and comments about the manuscript, which undoubtedly allow us to improve and enrich it. I list them and justify the changes made:

  • Line 131: I modify the topographic error (non-small cell lung cancer)
  • 141 – 144: I rewrite the sentence to make it less confusing.
  • Line 68-69: We revised the reference provided and being that the paragraph summarizes many concepts explained in the article, we added this sentence to complete the information about CTLA-4 and index the reference to the manuscript.

https://www.sciencedirect.com/science/article/abs/pii/S1471489218300031?via%3Dihub

  • Line 98-103 and 107-112: We complete the background TME being more exhaustive in the explanation about the interaction between innate immune cells and cancer cells.

    https://www.ncbi.nlm.nih.gov/pmc/articles/PMC6801651/

  • Line 434- 445: We revised the articles provided and introduce this parragraph about immunosenescence, trying to highlight its special characteristics as well as the most important discoveries discussed in the articles.

    https://www.ncbi.nlm.nih.gov/pmc/articles/PMC5917671/

    https://www.ncbi.nlm.nih.gov/pmc/articles/PMC6539213/

  • References are updated by adding the four articles provided.

Thank your very much for your time

Reviewer 2 Report

In this article, the authors perform an excellent review of the distribution of immune cell populations in both tumor environment and peripheral blood in patients with lung cancer and its importance as predictive biomarkers of efficacy for immune checkpoint therapies. In my opinion, this article can be accepted for publication in its present form.

Author Response

Thank you for your words and thank you very much for the time spent reviewing it. We are pleased that the review was to your liking.